



# TAMS: A Tracking, Classifying, and Variable-Assigning Algorithm for Mesoscale Convective Systems in Simulated and Satellite-Derived Datasets

Kelly M. Núñez Ocasio[1] and Zachary L. Moon[1,2]

[1]NSF National Center for Atmospheric Research, Boulder, CO, USA
[2]Earth Resources Technology (ERT), Inc. Laurel, MD, USA

**Correspondence:** Kelly M. Núñez Ocasio (knocasio@ucar.edu)

**Abstract.** The Tracking Algorithm for Mesoscale Convective Systems (TAMS) is a tracking, classifying, and variable-assigning algorithm for mesoscale convective systems (MCSs). TAMS was initially developed to analyze MCSs over Africa and their relation to African easterly waves using satellite-derived datasets. This paper describes TAMS v2.0, an open-source MCS tracking and classifying Python-based package that can be used to study both observed and simulated MCSs. Each step of the

5 algorithm is described with examples showing how to make use of visualization and post-processing tools within the package. A unique and valuable feature of this MCS tracker is its support for unstructured grids in the MCS identification stage and grid-independent tracking of MCSs, enabling application across various native modeling grids and satellite-derived products. A description of the available settings and helper functions is also provided. Finally, we share some of the current development goals for TAMS.

## 10 1 Introduction

Robustness in statistical analysis of mesoscale convective systems (MCSs) data has been possible in large part due to objective tracking methods. Automated tracking algorithms became more feasible with the advent of satellites in the 1980s. These algorithms, which initially made use of infrared (IR) or a combination of IR and visible images, enabled the detection and tracking of MCSs globally (Williams and Houze, 1987; Velasco and Fritsch, 1987; Augustine and Howard, 1991; Laing and

15 Fritsch, 1993; Machado et al., 1998; Mathon and Laurent, 2001; Vila et al., 2008). Testing has been conducted on brightness temperature ($T_b$) thresholds, which range from 253 K for the highest brightness temperature associated with convection to 213 K for very deep convection, to detect convective cloud areas (Maddox, 1980; Mapes and Houze, 1993; Machado et al., 1998; Goyens et al., 2012). In the context of tracking, the overlapping technique, which is still widely used today, relies on consecutive satellite image overlaps to establish time continuity (Williams and Houze, 1987; Evans and Shemo, 1996; Mathon

20 and Laurent, 2001). Since then, other tracking methods have been developed, including those based on the minimization of a cost function related to the speed and direction of the MCS (Hodges, 1995), the assessment of the magnitude of spatial correlation (Carvalho and Jones, 2001), considering the system's propagation speed (Woodley et al., 1980), and employing a projecting centroid technique for forecasting the feature's location in the current or next time step (Johnson et al., 1998).





Radar data have also played a crucial role in facilitating the nowcasting and forecasting of MCSs enabling the detection of
storm cells using radar volume scan data. This data is used to match storms across scans and forecast their position, either based
on the storm centroid or through cross-correlation methods that utilize 2-D reflectivity data to calculate motion vectors (Johnson
et al., 1998; Dixon and Wiener, 1993). At present, we observe the use of automated methods that leverage these techniques,
detection methods, or a combination thereof, along with other satellite-derived products and methods. In the review that follows,
our focus is on recent MCS and storm-tracking algorithms that are currently available as open-source code or packages,
facilitating ongoing research on MCSs and promoting open and inclusive research, in line with the strong encouragement from
the international scientific community.

The "Grab 'em Tag 'em, Graph 'em" (GTG) algorithm (Whitehall et al., 2015) (https://kwhitehall.github.io/grab-tag-graph/,
last access: 29 October 2023) identifies cloud clusters using IR and a corresponding graph node to track via the area over-
lap method. GTG is a Python-based tracker, and it can be applied to remote-sensed datasets. The Tracking and Object-Based
Analysis of Clouds (tobac; Heikenfeld et al., 2019; Sokolowsky et al., 2023) (https://github.com/tobac-project/tobac, last ac-
cess: 29 October 2023) is a community-developed Python package with three main modules: detection, segmentation, and
linking. In contrast to the area overlap, tobac assigns a center point to a track when it falls within a predicted radius of
motion, which is done using trackpy (http://soft-matter.github.io/trackpy). Unlike area-overlap algorithms where the feature
geographical area is treated as the segmentation area and is not separated from the identified feature, in tobac, the orig-
inal identified feature can be different from the segmentation area, which is calculated with a watershed method. Similar
to tobac is the TempestExtremes (Ullrich et al., 2021)(https://github.com/ClimateGlobalChange/tempestextremes) algorithm
which is a flexible tool box in which the user can choose to track different extreme events not limited to MCSs. The Python
FLEXible object TRacKeR (PyFLEXTRKR; Feng et al., 2023) (https://github.com/FlexTRKR/PyFLEXTRKR, last access:
October 2023) uses both IR and surface precipitation to identify and track convective features using the renown area-
overlap method. The PyFLEXTRKR includes several multi-object identification algorithms to track from cells to MCSs and
can explicitly handle merging and splitting. It also has a 2-D convective cell advection estimate functionality specific for re-
flectivity data using preexisting Python packages. The Multi-Object Analysis of Atmospheric Phenomenon (MOAAP; Prein
et al., 2021, 2023) (https://github.com/AndreasPrein/AndreasPrein-MCS-tracker-intercomparison-SAAG, last access: 30 Oc-
tober 2023) algorithm is a Python-based MCS tracker that uses $T_b$ to connect convective features via preexisting image-
processing tools rather than using an overlap threshold. MOAAP can also be used to track other atmospheric features. Of
the trackers reviewed, PyFLEXTRKR and tobac offer parallelization capability for at least one step of the algorithm. Lastly,
convective-cell trackers specifically for radar volume scans that are available as open-source codes are the Thunderstorm Iden-
tification, Tracking, Analysis, and Nowcasting (TITAN; Dixon and Wiener, 1993) (https://github.com/NCAR/lrose-titan, last
access: 30 October 2023) and similarly TINT (Raut et al., 2021) (https://github.com/openradar/TINT, last access: 30 October
2023).

Two tracking algorithms, the Forecasting and Tracking the evolution of Cloud Clusters (ForTraCC; Machado et al., 1998)
and the Tracking Of Organized Convection Algorithm (TOOCAN; Fiolleau and Roca, 2013), are not publicly available. These
trackers are reviewed here because their development has provided and continues to provide publicly available data on MCSs



over South America (ForTraCC) and globally (TOOCAN). ForTraCC is currently employed operationally for nowcasting at the Brazilian Center for Forecast and Climate Studies of the National Institute of Spatial Research. It can utilize input data from radar, precipitation, or outgoing longwave radiation and employs an area-overlapping tracking approach. TOOCAN identifies the most convective cores of the cloud top, which are characterized by the lowest brightness temperature. Using clustering methods, it associates these cloud clusters over time. This approach enables the study of MCSs by decomposing them into their numerous small-scale features, facilitating earlier detection in their life cycle.

## 1.1 TAMS v1.0

In this paper, we will describe the new open-source Python-based tracker known as the Tracking Algorithm for Mesoscale Convective Systems (TAMS). However, before discussing TAMS, we will review the first version ("TAMS v1.0"). TAMS was initially developed to track and analyze tropical MCSs over Africa associated with African easterly waves (Núñez Ocasio et al., 2020a, b). These MCSs are directly associated with the formation of tropical cyclones in the Atlantic (Núñez Ocasio et al., 2020b; Rajasree et al., 2023), the intensity of the West African Monsoon (Núñez Ocasio et al., 2021), and initiation by topography (Hamilton et al., 2020, 2017), which all together, adds layers of complexity to study them. This first version of TAMS comprised four main steps: identification, tracking, classification, and the assignment of precipitation. A cloud element (CE) identified as a potential MCS candidate had to meet specific criteria, including a $T_b$ area of 235 K or lower, with an area $\geq$ 4000 km$^2$ of embedded 219 K $T_b$ regions (cold cores). The tracking step involved a combination of the area-overlap technique and a modified version of the centroid-projection technique. In this modified approach, with the longitudinal component of the CE centroid, the CE was projected using a fixed climatological zonal propagation speed, for which projected distance depended on the temporal resolution of the data. In Núñez Ocasio et al. (2020a), the CEs were projected two hours into the future. If CEs of the current time sufficiently overlapped (i.e., 55%) with CEs of the next satellite image (and thus, forward linking), then the future CE was flagged as a 'kid' of the current satellite image CE 'parent'. The flagged 'families' of CEs were then grouped (the actual track) using a recursive graph-walking function. The entire track of an MCS was then classified into one MCS class following specific area, shape, and time criteria (see Núñez Ocasio et al. (2020b), their Table 1). There were four main classes defined in TAMS v1.0 comprising two main group: organized and disorganized systems. The four specific classes were: Mesoscale Convective Complexes (MCCs), Convective Cloud Clusters (CCCs), Disorganized Long-Lived (DLLs), and Disorganized Short-Lived (DSLs). The last step of TAMS v1.0 was the assignment of precipitation, which used IMERG data interpolated for compatibility with the 3-km IR data. At the time, regridding from higher-resolution precipitation estimates, such as IMERG, rather than coarser TRMM 3B42 estimates, represented a significant improvement in TAMS compared to other MCS tracking algorithms.

TAMS v1.0 allowed Núñez Ocasio et al. (2020a) to analyze the morphology and climatology of MCSs Africa, a region with various spatial- and time-scale atmospheric phenomena that interact (e.g., West African Monsoon, African easterly waves (AEWs), African easterly jet (AEJ), and the ITCZ), especially over the northern summer months. It was found that a realistic representation of MCS propagation over Africa is dependent on the AEJ, and realistic MCS tracks are only attained when the tracking technique accounts for the AEJ mean background flow through cloud projection (Núñez Ocasio et al., 2020a). These



results were important as they showed that tracking technique differences can bias the lifetimes of longer-lived convective systems that, over Africa during the northern summer, tend to be coupled to AEWs. The projection of cloud edges using a climatological MCS propagation speed facilitated the area-overlap method by enhancing the probability of connecting cloud elements (CEs) that belonged to the same system. Additionally, this projection approach also facilitated the handling of splits and mergers.

TAMS v1.0 was also applied to a study that investigated MCSs coupled to AEWs. With a combination of MCS tracks using TAMS v1.0 and AEW tracks (Brammer et al., 2018), a wave-relative framework was developed in order to study the complex interactions between MCSs and AEWs. Núñez Ocasio et al. (2020b) found that there are significant differences between the MCSs of AEWs that become tropical cyclones (TCs) and the MCSs of AEWs that do not undergo TC genesis. MCSs of developing AEWs (those that become TCs) are more likely to be of the squall line type over Africa and the Eastern Atlantic. These MCSs are more likely to move at the same speed and positioned at the wave vortex they are coupled to.

TAMS v1.0 was written in MATLAB, and it was not an open-source code. Unlike some other trackers, it did not use a preexisting package to track or link CEs detected; rather, the code was written entirely from scratch using some MATLAB image processing. The objective of this paper is to introduce and describe the new "TAMS v2.0" Python-based open-source project. TAMS v2.0 retains the essence of TAMS V1.0 while introducing new capabilities, configurations, and settings to make it more user-friendly and flexible. It also provides visualization and post-processing tools.

The following sections are organized as follows: Section 2 will provide an overall description of TAMS v2.0 with subsections that detail each of the algorithm's steps and related utilities. Section 3 will focus on introducing additional settings and helper functions as part of the software. Section 4 will be a summary, including a subsection for the caveats and development goals.

## 2    TAMS v2.0 Overall Description

The Tracking Algorithm for Mesoscale Convective Systems (TAMS) v2.0 is an open-source MCS tracking and classifying Python-based package. A standard workflow using TAMS v2.0 follows the same main four steps as its predecessor: 1) Identify, 2) Track, 3) Classify, and 4) Assign variable(s) (Figure 1), but the steps can also, to some extent, be run independently. TAMS v2.0 now includes the capability to assign any desired variable or atmospheric field of choice (not limited to precipitation as in TAMS v1.0) to each mesoscale convective system (MCS) for calculating corresponding statistics of the variable within the cloud area. This new functionality, which will be discussed in detail in the following sections, allows TAMS v2.0 to be customized to identify and track using additional criteria specified by the user, rather than solely relying on $T_b$-dependent criteria.

The TAMS documentation page (https://tams.readthedocs.io/) provides a set of instructions for Conda/Mamba or pip installation methods. The package includes a set of visualization, sample datasets, and post-processing tools including functionalities that will be discussed in the following sections. One unique functionality of TAMS v2.0 is that, like TAMS v1.0, it is grid-independent for tracking, making it a valuable tool for validating simulated MCSs against observed systems. TAMS v2.0 is additionally grid-independent for assigning variables and supports unstructured grid input in the identification stage. TAMS





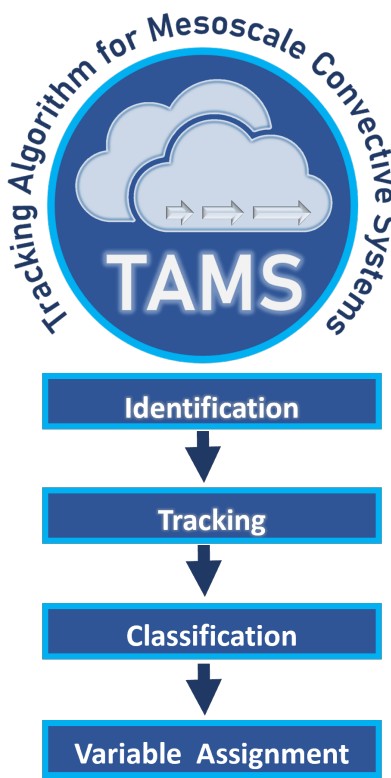

**Figure 1.** Schematic of TAMS logo and the four main algorithm steps.

v2.0 provides optional parallelization for (1) CE identification and (2) calculation of statistics of gridded data within MCS and cold-core areas. TAMS v2.0's core dependencies are: GeoPandas (den Bossche et al., 2023), Matplotlib (Hunter, 2007), NumPy (Harris et al., 2020), pandas (Wes McKinney, 2010; The pandas development team, 2023), scikit-image (van der Walt et al., 2014), Shapely (Gillies et al., 2023), and xarray (Hoyer and Hamman, 2017). The documentation examples are provided
as Jupyter Notebooks. The output format is a GeoPandas GeoDataFrame, but this can be transformed into a gridded mask representation, as done for the Prein et al. (2024) tracker intercomparison project (code available in TAMS GitHub). Next, each step of the algorithm will be discussed in detail. From hereon, TAMS v2.0 will be addressed as TAMS.

## 2.1 Identification

The default TAMS identification follows the approach of the first version, using contours of $T_b$ or cloud-top temperature data (if
using model data) to identify cloud regions (i.e., CEs) of 235 K areas containing embedded 219 K area(s) $\geq 4000$ km$^2$. The field used must have associated latitude and longitude coordinate variables. Extremely small 219 K areas ($\leq 10$ km$^2$) are discarded, as well as 235 K areas that do not meet the 4000 km$^2$ threshold. These contours definitions are converted to Shapely polygons, with the convex hull operation applied in order to smooth out the shapes and reduce the number points required to represent



them (to reduce memory requirements and speed up computations). The `tams.identify` function returns a GeoDataFrame of these shapes. This is the first step of the TAMS workflow and it can also be ran within the `tams.run` function which runs all the steps of the workflow with one command (note that if using `tams.run`, the input xarray DataArray should have both cloud-top temperature and precipitation rate variables).

Other parameters in this identification step include enabling parallelization (for input data with a time dimension), `ctt_threshold` (cloud-top temperature threshold for cloud boundary) and the `ctt_core_threshold` used to identify the deep convective areas and embedded cold cores, respectively. Figure 2 is a visual example using the Meteosat Second Generation (MSG) geostationary satellite (Schmetz et al., 2002), specifically the 10.8 $\mu$m spectral band of the Spinning Enhanced Visible and Infrared Imager (SEVIRI) data on September 16, 2022. MCSs coupled to an AEW that were active during research flight seven of field campaign CPEX-CV (https://espo.nasa.gov/cpex-cv/content/CPEX-CV, last access: 31 October 2023) were identified. The red crosses show the track of the AEW up to that time using AEW tracker by Lawton et al. (2022). The TAMS package includes sample MSG satellite data that can be loaded as either the satellite IR radiance (channel 9) using the `tams.data.load_example_ir` function or the `tams.data.load_example_tb` function which loads derived $T_b$. The user can compute the $T_b$ by using function `tams.data.tb_from_ir` with MSG SEVIRI IR satellite radiance as input (https://www.eumetsat.int/media/8571).

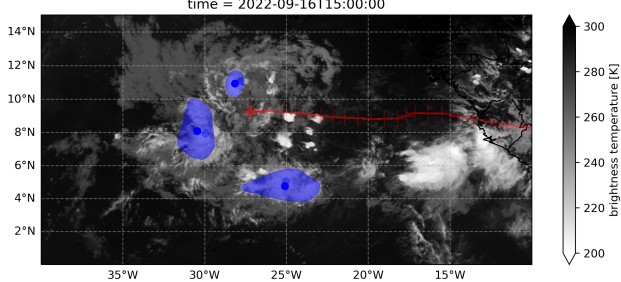

**Figure 2.** Identified MCSs using MSG data during research flight seven of CPEX-CV field campaign. CEs (convex-hull, warm-core 235-K areas) are in blue and IR in shade. Blue dots represent geographical centroid position. Red crosses are the track of the AEW. Time is in UTC.

### 2.1.1 Grid-Independent Identification

TAMS includes sample data taken from a 15-km global-mesh MPAS simulation from 2006-09-08 12:00 UTC to 2006-09-13 18:00 UTC initialized with IFS Núñez Ocasio and Rios-Berrios (2023, 2022). This includes native unstructured-grid model output and output regridded to a 0.25° lat/lon grid using a first-order conservative method Jones (1999) via CDO. To keep file sizes down, the versions included in TAMS are spatial subsets of the original model outputs and include only selected variables. The sample datasets can be automatically downloaded with `tams.data.download_examples` and then loaded using `tams.load_example_mpas` and `tams.load_example_mpas_ug`, respectively.



TAMS is able to identify CEs in both MPAS datasets by contouring the Cloud-top temperature (CTT) threshold. For unstructured-grid data, this currently uses Matplotlib's `tricontourf` function. By default the Delauney triangulation of the lat/lon coordinates is used. Note that this is only an approximation of the true MPAS grid, which is mostly made up of hexagons, but we expect this effect to be insignificant for our high-resolution example. MPAS grids can be decomposed into
triangles (e.g. 6 per hexagon) and the resulting triangulation can be used instead if desired (https://mpas-dev.github.io/MPAS-Tools/stable/visualization.html#mpas-mesh-to-triangles) for a faithful representation but with increased contouring compute time.

Figure 3 shows a comparison of MPAS unstructured grid identified CEs compared to those identified from the MPAS data.

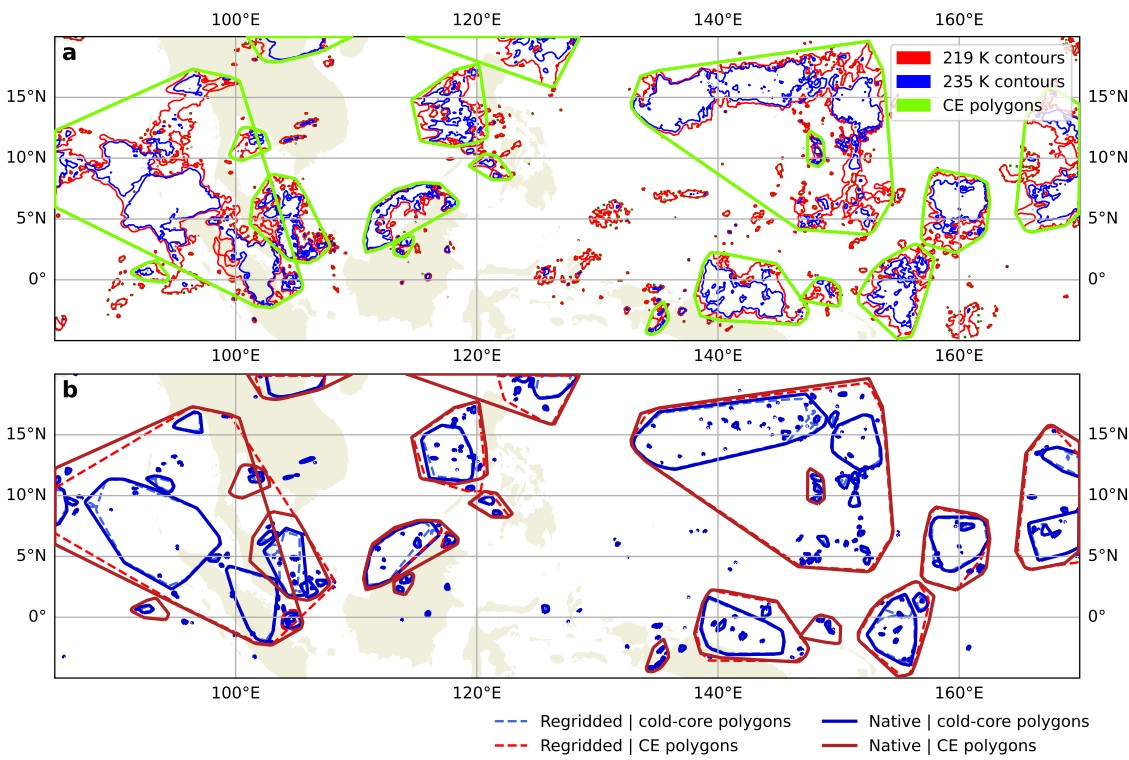

**Figure 3.** CEs identified at 2006-09-08 22:00 UTC (10 hours in) for the sample MPAS datasets. a) contours of the native MPAS output and associated CE polygons. b) CEs derived from native (solid lines) MPAS output and regridded (dashed lines) MPAS output.

## 2.2 Tracking

By default, the track of an MCS is obtained by linking CEs of the current CTT to CEs of the previous CTT (backward linking) based on maximum CE polygon overlap (`tams.overlap`). A CE at time $t_i$ has one 'parent' at $t_{i-1}$, but multiple CEs can have the same parent (e.g. splitting). Optionally, you can choose to have only the largest CE continue a given track as done in Evans and Shemo (1996). Forward linking is also available, and other linking methods are in development.





Overlap thresholds are based on normalized area $(0 - -1)$, so it is important to consider which area to use when normalizing
the overlap area. Núñez Ocasio et al. (2020a) used the minimum area between the two CEs being compared. In this version of
TAMS, by default, the area of the CE at the current time when linking is used, but `tams.overlap` provides other options: the
minimum CE area, maximum CE area, the average, the current CE, or the CE at the other time. As in the first version, there is
a setting to project the CE (`tams.project` function) in the $x$ direction before computing overlap, which proves useful when
MCSs are propagating in an environment with prevalent background flow, especially when hourly or higher-time-resolution
data is not available (Núñez Ocasio et al., 2020a; Feng et al., 2023). The track of an MCS is then defined as the record of all
CEs linked. It is important to note that for a given MCS (ID), there may be more than one CE at a given time. To define a single
point-wise track in this case, we use the centroid of the combined 'multi-polygon' at each time.

Figure 4 shows MCSs identified and tracked through time (darker shades represent later times) from the sample MSG
satellite data using the `tams.plot_tracked` function to plot the tracks. How different setting combinations can be used
to link and subsequently, alter the track of an MCS is shown in Table 1 for the MCS in the red box in Figure 4. In order to
capture the full track of an MCS that evolved from a larger area to a smaller area as in the example, projection and overlap
normalization settings should be carefully selected. Only the two cases with $u$ projection -15 m s$^{-1}$ and average or minimum
overlap normalization continue this track.

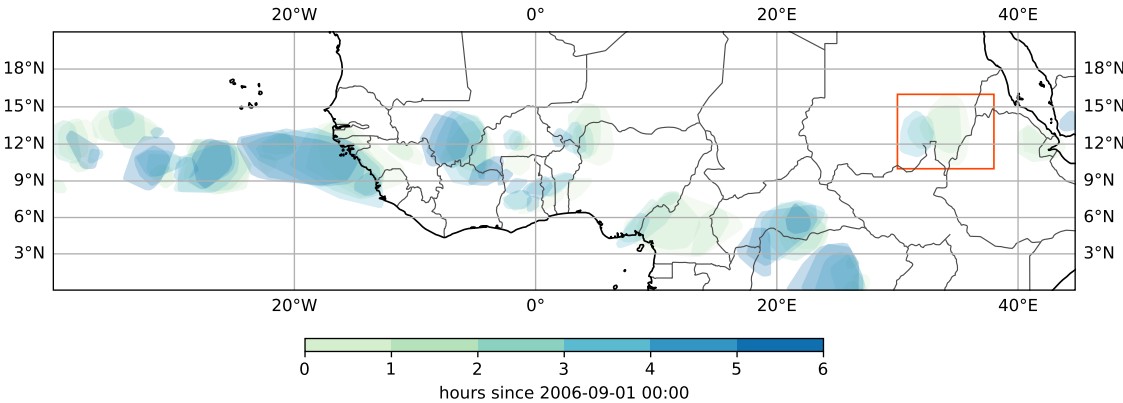

**Figure 4.** MCS tracks in time (darker shades represent later times) from sample MSG satellite data.

A variable that can be analyzed strictly based on the information from the CE and MCS shapes is the area of the systems.
Figure 5 shows how the area of simulated MCSs in MPAS is sensitive to threshold temperature selection for identification and
to projection speed for tracking. The area is more sensitivity to temperature tuning than projection speed, since there is little
noticeable change between the distributions within a certain column. For example, the most likely area range is around $10^5$
km$^2$ for all cases in the first column. However, higher projection speed results in a larger number of MCSs in all temperate
threshold cases, suggesting that less linking is occurring. Warmer temperature thresholds compared to the defaults (rightmost
column) lead to increased system counts and a smoother distribution covering a broader area range. A warmer threshold leads
to mergers in the contouring, but also provides more new shapes (in this case the latter wins out). On the other hand, much





**Table 1.** Example using different tracking options for an observed MCS case.

| case | $u$ proj [m s$^{-1}$] | linking | overlap norm | continued? |
|---|---|---|---|---|
| 0 | 0 | backward | average | no |
| 1 | 0 | forward | minimum | no |
| 2 | 0 | forward | maximum | no |
| 3 | -15 | backward | average | yes |
| 4 | -15 | forward | minimum | yes |
| 5 | -15 | forward | maximum | no |

cooler temperature thresholds (left column) pick up about half as many systems, with a distribution that favors larger areas. In this case, the cold-core area criterion is more restrictive.

Another way in which the area of MCSs can be analyzed is by comparing it with other MCS characteristics such as duration. Figure 6 shows duration versus area 2-D KDEs for simulated MCSs from MPAS sample data. In this example we see that the systems with the longest duration are not necessarily the largest. Here, except for some outliers, the systems with the longest duration are here of low–moderate area, while the largest systems are of low–moderate duration. Moreover, there are different relationship between area and duration between different core thresholds (e.g., the slope is steeper for the colder cases).



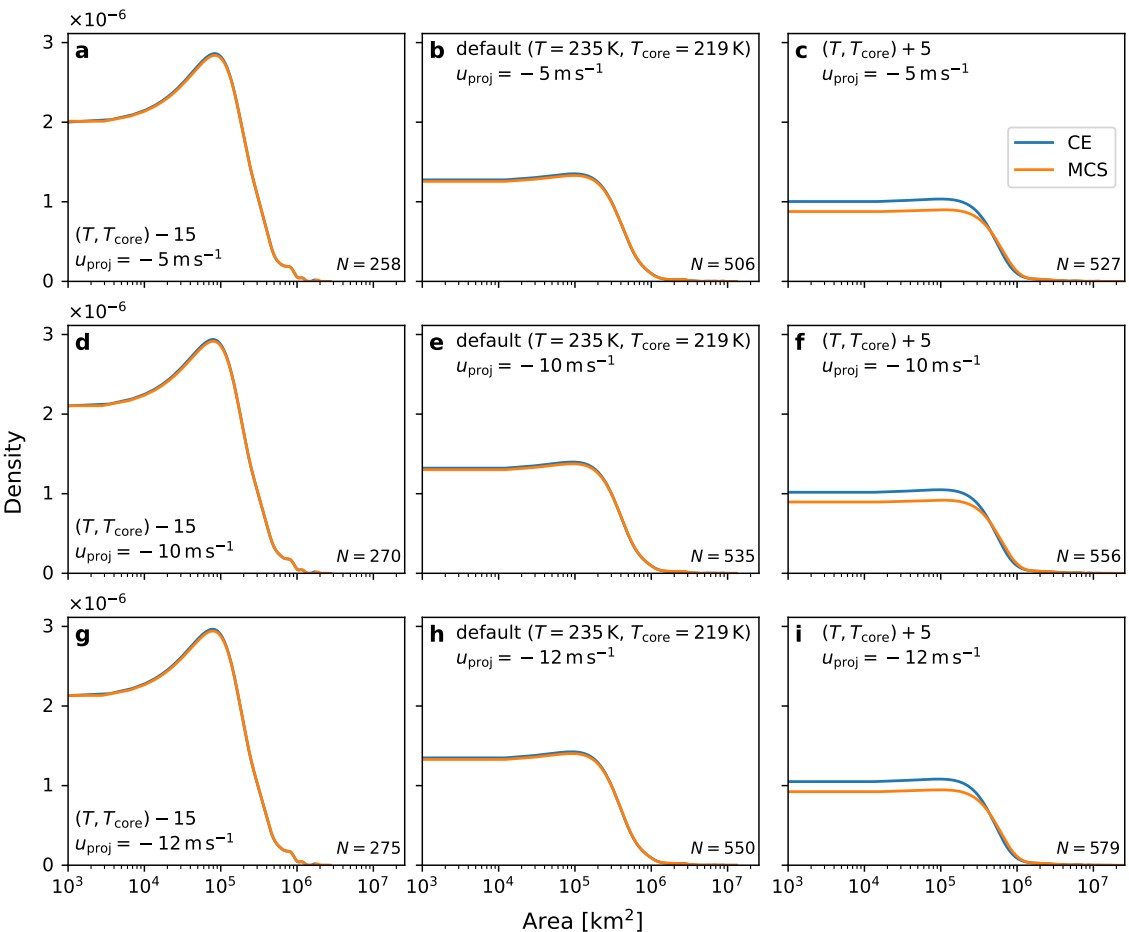

**Figure 5.** Maximum area kernel density estimates (KDEs) for simulated MCSs from MPAS sample data interpolated to a latitude/longitude grid. Rows represent results using the same projection speed for tracking while columns represent different temperature thresholds for identification. The data is for the period of 2006-09-08 12:00 UTC to 2006-09-13 18:00 UTC.

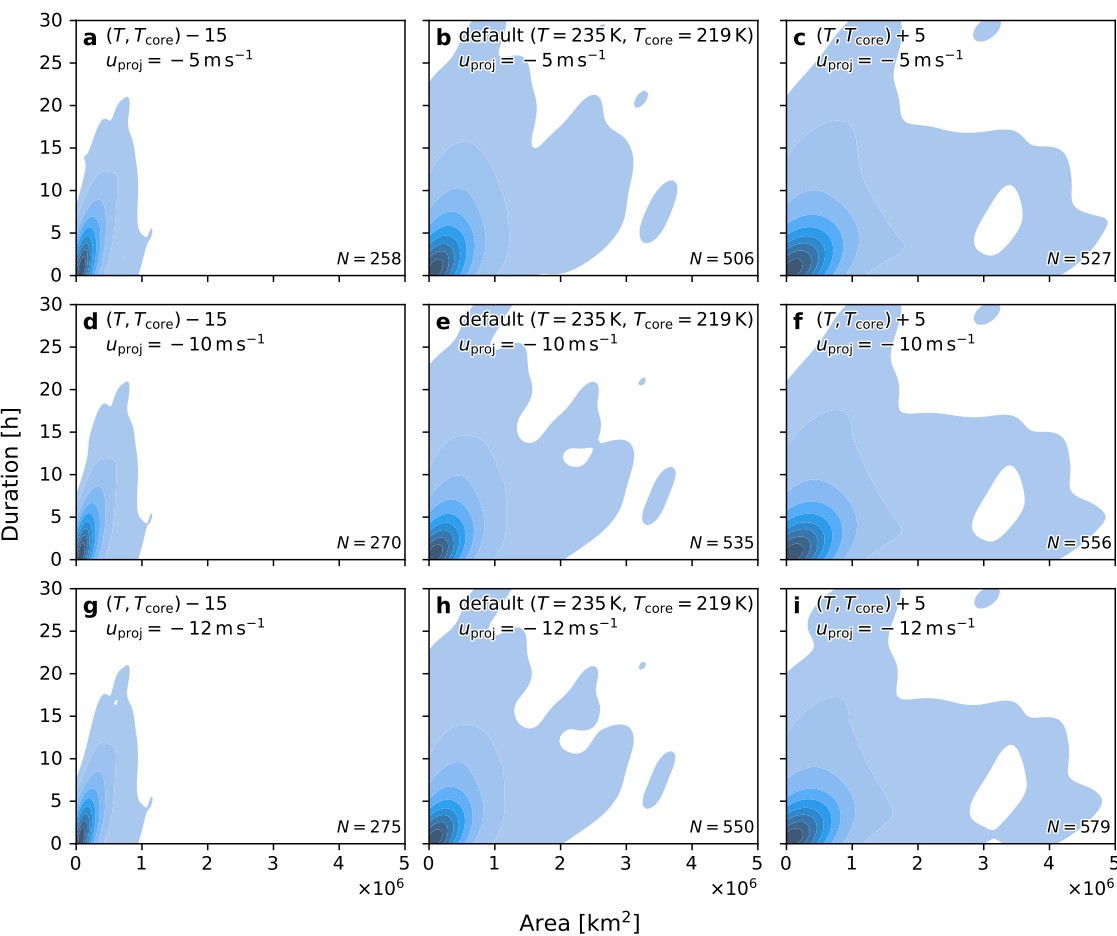

**Figure 6.** Duration versus maximum area 2-D kernel density estimates (KDEs) for simulated MCSs from MPAS sample data interpolated to a latitude/longitude grid. Rows represent results using the same projection speed for tracking while columns represent different temperature thresholds for identification. The data is for the period of 2006-09-08 12:00 UTC to 2006-09-13 18:00 UTC.





## 2.3 Classification

The TAMS software allows for the user to classify the entire lifetime of an MCS into one class based on the area, shape, and time criteria described in Table 2. There are four main classes defined in TAMS within two main groups: organized and disorganized systems. The four specific classes are: Mesoscale Convective Complexes (MCCs), Convective Cloud Clusters (CCCs), Disorganized Long-Lived (DLLs), and Disorganized Short-Lived (DSLs). These criteria mostly follow those of the first version in Núñez Ocasio et al. (2020a) with the distinction that DSLs are classified as anything with duration shorter than

6 hours (rather than 3 hours or less). Note that this criteria need not to be met for consecutive times but for at least for 6 hours within the lifetime of the system. Recall that all TAMS CEs meet the cold-core criterion.

**Table 2.** Criteria to Categorize the four Classes of MCSs

| Organized Systems | Disorganized Systems |
|---|---|
| *Mesoscale Convective Complex (MCC)* | *Disorganized Long-Lived (DLL)* |
| Size: < 219 K region has area ≥ 25,000 km$^2$ | Temperature: < 219 K |
| < 235 K region has area ≥ 50,000 km$^2$ | Duration: Sustains for ≥ 6 hours |
| Duration: Size definitions are met for ≥ 6 hours | No size or shape criterion |
| Shape: $\varepsilon = \sqrt{1 - \frac{b^2}{a^2}} \leq 0.7$ | |
| *Convective Cloud Cluster (CCC)* | *Disorganized Short-Lived (DSL)* |
| Size: < 219 K region has area ≥ 25,000 km$^2$ | Temperature: < 219 K |
| Duration: Size definitions are met for ≥ 6 hours | Duration: Sustains for ≤ 6 hours |
| Shape: No shape criterion | No size or shape criterion |

In order for each MCS to be classified, the user can input the GeoDataFrame obtained from running `tams.track` into the `tams.classify` function. The result will be a GeoDataFrame with an additional categorical `mcs_class` column added to the input frame. This GeoDataFrame retains information associated with each CE, and, in addition to the ttmcd_id column, it

allows for analyzing the number of CEs versus the number of MCSs by category, as shown in Figure 7.

Over western Africa during the Figure 7 example period, the majority of the convection is classified as DSL. This prevalence of DSLs is notable, as a significant portion of the convection in this region belongs to this category (Figure 7a and b). Disorganized convective systems are known to be the most frequent type of convective systems in the tropics (Rossow et al., 2013; Tan et al., 2013; Semunegus et al., 2017; Núñez Ocasio et al., 2020a). Furthermore, the number of organized MCSs (i.e., MCSs and

220 CCCs) is greatly influenced by the ITCZ, West African Monsoon, and AEWs, particularly during the northern summer months (Berry and Thorncroft, 2005; Núñez Ocasio et al., 2020b, 2021; Núñez Ocasio and Rios-Berrios, 2023).

As expected, the number of CEs is much greater than the count of MCSs. (An MCS only gets counted once for its lifetime, while CE counts include counts at each time in the period.) Organized systems (MCSs and CCCs) were less common, with their count below 100. CCCs, which include squall-line type convection, were the second most prominent category during this

time. The number of DSL MCSs is comparable to the number of DSL CEs due to the shorter MCS durations. Duration box





plots, as shown in Figure 7c, can help the user discern differences in the duration of the systems across categories. Although CCCs and MCSs share the same time criterion, it is evident that over September 2022, MCCs, based on the median, lasted longer than CCCs.

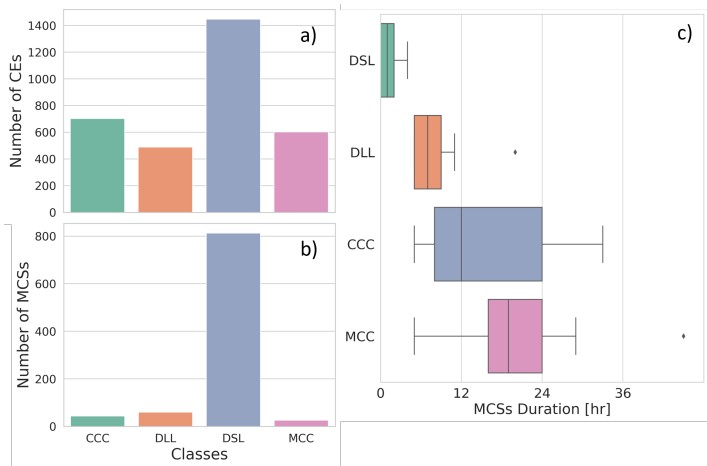

**Figure 7.** Example statistics of observed number of CEs (a) and MCSs (b), and duration of MCSs by category (c) during September 4th through the 30th, 2002, period of CPEX-CV field campaign, over a western Africa domain.

Figure 8 displays MCS area distributions from the CPEX-CV field campaign, categorized by their respective types. During
this period, disorganized systems (DSLs and DLLs) were generally smaller than organized systems (MCCs and CCCs), with MCCs, as expected, having the largest median area.

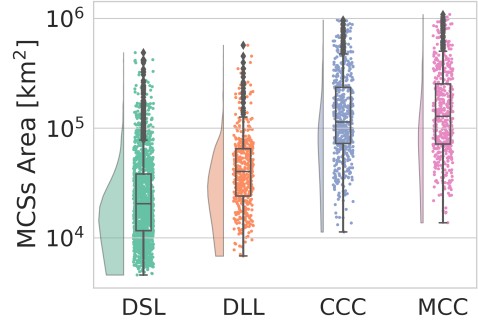

**Figure 8.** Raincloud plots of MCS area (note the log scale) by category for the same period as in Figure 7.

## 2.4 Variable Assignment

Another unique functionality of TAMS, in addition to identifying and tracking systems using CE polygons and being grid-independent, is its ability to assign any variable of choice to CE or MCS shapes and compute related statistics on the data



within using the `tams.data_in_contours` function. This allows for users to assess the evolution, characteristics, and
nature of the systems under study using different atmospheric variables. This function is not limited to a specific data type,
spatial resolution, or a specific grid, as demonstrated in the previous sections. Same temporal resolution, or at least data that
includes the same times allows the user to match variables to each MCS across various datasets.

One common example is assigning precipitation to each MCS from satellite-derived products like IMERG, as shown in
Figure 9a for the example period of observed MCSs. DSLs, which had the greatest number of counts, were the rainiest type of
MCSs. However, the precipitation rate median of CCCs and MCCs were slightly above that of DSLs, at about 3 mm h$^{-1}$.

Through this function, the user can also extract the minimum $T_b$ as a diagnostic for convection intensity. In Figure 9b, the
most intense and coldest cloud tops were observed in MCCs during the example period. If the user or researcher is analyzing
model data that provides a full array of 2-D and 3-D variables at each grid cell, they are not limited to just cloud-top temperature
or precipitation. With this functionality, they can assign many other variables and compute relevant statistics.

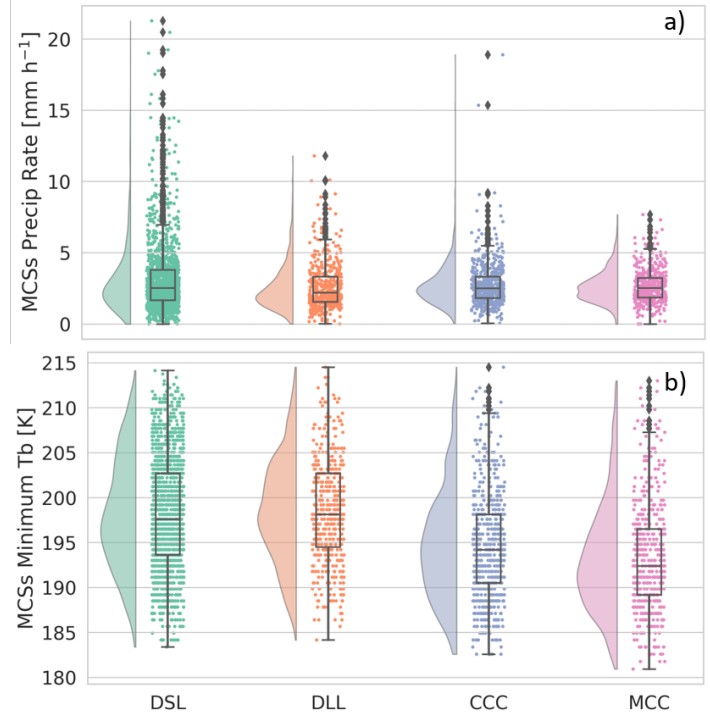

**Figure 9.** Raincloud plots of MCS minimum $T_b$ and area-mean rain rates by category for the same period as in Figure 7.

As another example, Figure 10 shows 2-D distributions of simulated precipitation rate versus minimum $T_b$ across different
setting options. For this particular subset of data, there is no distinct relationship between minimum $T_b$ and average precipitation
rates. Most of the differences across the setting options are due to sensitivity to the change in temperature threshold for
identification than to the selected projection speed as discussed in Section 2.2. The sample simulated precipitation rate dataset
is available in the package and can be loaded via `tams.load_example_mpas`.

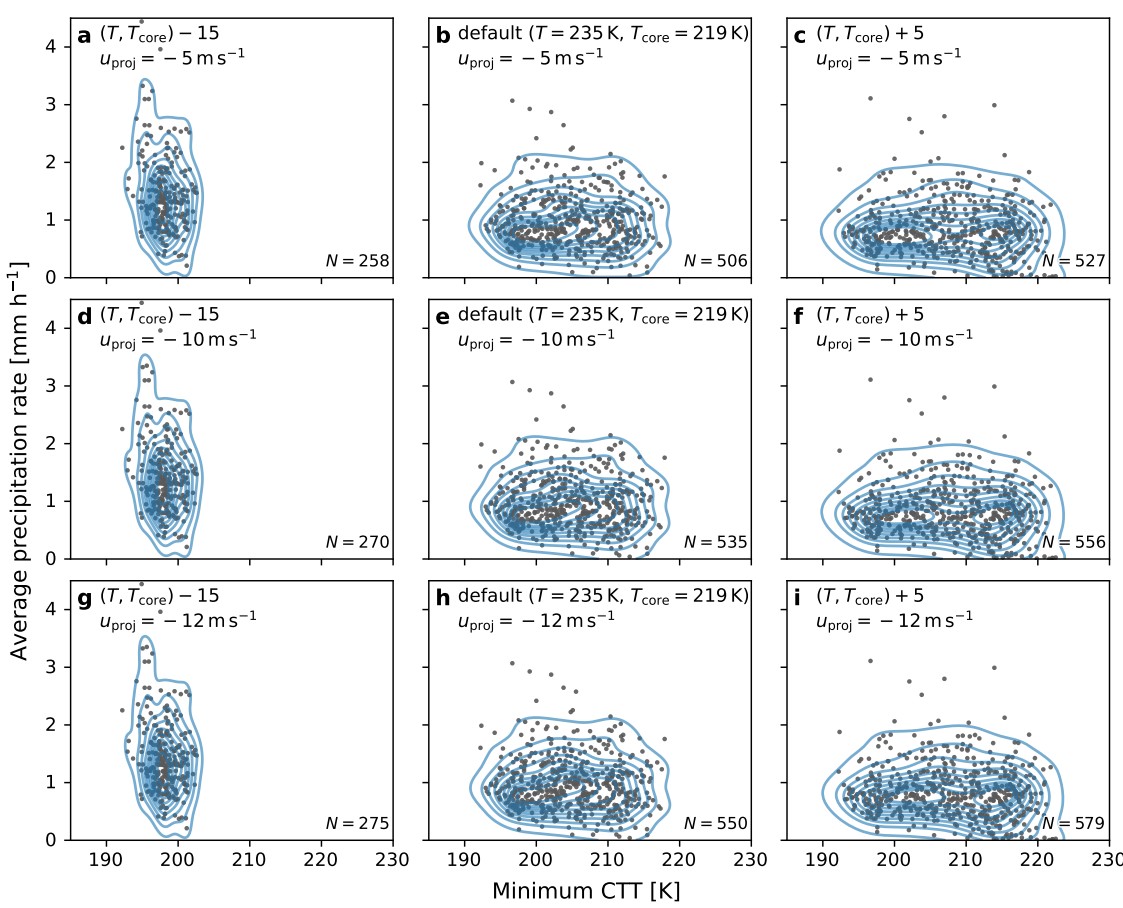

**Figure 10.** 2-D precipitation rate versus minimum $T_b$ kernel density estimates (KDEs) for simulated MCSs from MPAS sample data interpolated to a latitude/longitude grid. Rows represent results using same projection speeds for tracking while columns represent different temperature thresholds for identification. The data is for the period of 2006-09-08 12:00 UTC to 2006-09-13 18:00 UTC, as in Figure 5.



# 3 Additional Tools and Helper Functions

## 3.1 Eccentricity calculation

A quantity that can provide information about the structure of the system is eccentricity (Evans and Shemo, 1996; Núñez Ocasio et al., 2020a). The `tams.calc_ellipse_eccen` function can be used to calculate the first eccentricity ($\epsilon = \sqrt{1 - b^2/a^2}$) of the least-squares best-fit ellipse to the coordinates of the polygon's exterior. If the first eccentricity is $\leq 0.7$, the system is considered more circular (a circle has $\epsilon = 0$).

Figure 11 displays the evolution of eccentricity over time, along with other relevant variables, for an observed MCS identified and tracked during research flight eight of the CPEX-CV field campaign on September 20, 2022. This system persisted for four hours, and TAMS classified it as a DLL. Initially, the system exhibited a more elongated shape, reaching its peak precipitation rates one hour after initiation. As time progressed, the system gradually expanded and became more circular. By the time of dissipation, it had evolved into a large and nearly circular system, indicative of a dissipating anvil.

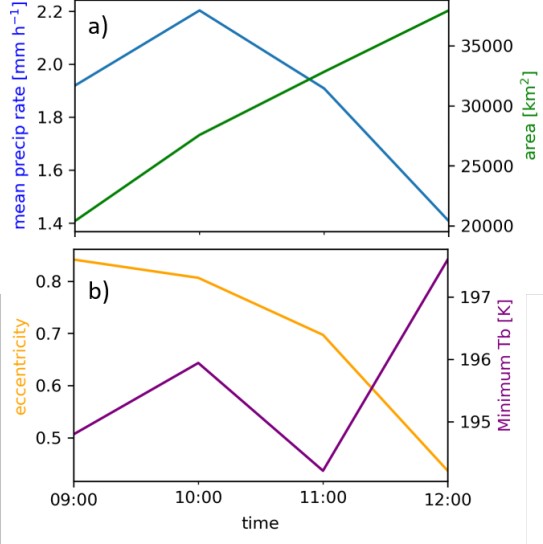

**Figure 11.** Time series of the observed MCS during research flight eight of the CPEX-CV field campaign of precipitation rate (blue line) and area (green line) (a), and eccentricity (yellow line) and minimum $T_b$ (orange line) (b).

## 3.2 Calculating cloud-top temperature from simulated OLR

TAMS also includes code to calculate the cloud top temperature from simulated top of the atmosphere OLR following the Stefan–Boltzmann law. This is used within `tams.load_mpas_precip`, the function used to load simulations nearing near-realtime application for the PRECIP field campaign. The Yang and Sligo (2001) method has also been implemented, and others are planned.



### 3.3 Post-processing and Visualization

Additional examples featuring the sample data and visualizations can be found within the documentation (https://tams.readthedocs.io/).
Example applications can be found in the examples directory of the GitHub repository. Post-processing and code for Figures 4,
5, 6, and 10 which show the different tracking and identification options, as well as the use of the `tams.load_example_mpas`
function, can be found in the documentation under examples. Similarly, the code for Figure 3, demonstrating tracking on
unstructured-grid data, is published in the documentation.

### 4  Summary

We have introduced and described TAMS, an open-source MCS tracking and classifying Python-based package available
through GitHub that can be used to study both observed and simulated MCSs. In addition to describing each of the core
steps of the algorithm (1) Identify, 2) Track, 3) Classify, and 4) Assign variable(s), we have introduced visualization and post-
processing tools within the package to facilitate the analysis of MCSs for users. What distinguishes TAMS from other tracking
packages is its specific design for tracking MCSs. However, the identification and tracking components are flexible and could
easily be applied to variables other than cloud-top temperature for different tracking situations. The identification of MCSs
is grid-independent, and the user can assign any desired variable to compute statistics on it within the MCS area over time.
Current settings and helper functions enable various ways to calculate overlap and account for background flow. The user can
select the preferred threshold temperature for the larger convective area to be detected and tracked, as well as the threshold for
the corresponding cold cores. Options to select the desired linking method (backward or forward, keep largest, etc.) are also
available. Through the sample data available, we have shown that TAMS works with satellite and model data, as well as with
unstructured-grid native MPAS data.

Recently, TAMS was applied as an MCS forecasting tool using MPAS near-real-time forecast in recent PRECIP (http://precip.org/,
last access: 12 December 2023) and CPEX-CV (https://ghrc.nsstc.nasa.gov/home/field-campaigns/cpex-cv, last access: 12 De-
cember 2023) field campaigns. In addition, TAMS is one of six MCS tracking algorithms participating in a multi-MCS-tracking
intercomparison study (Prein et al., 2024), which is part of the NSF NCAR South America Affinity Group (SAAG; Dominguez
and Co-authors, 2023)(https://ral.ucar.edu/projects/south-america-affinity-group-saag, last access: 12 December 2023) to study
MCSs over South America. The specific TAMS SAAG application can be located under `examples/mosa` in the GitHub
repository. PyFLEXTRKR, MOAPP, ForTraCC, tobac, and TOOCAN are also involved in this study.

TAMS is being used internationally for MCS detection and tracking. Currently, TAMS is part of a second multi-MCS-
tracking intercomparison study that is making use of high-resolution global simulations from DYAMOND (https://www.esiwace.eu/the-
project/past-phases/dyamond-initiative, last access: 12 December 2023) to analyze MCSs globally. We hope TAMS can serve
the scientific and research community worldwide, facilitating the study of observed and simulated MCSs. We very much
welcome contributors to the code.



## 4.1 Caveats

There are a couple of caveats associated with the software that are worth noting. The use of convex hull to define CEs introduces some systematic bias towards larger areas. This could potentially lead to an artificial increase in the calculated system's propagation speed relative to the geographical centroid. More generally, however, computing overlap with convex hull shapes allows for larger likelihood of linking compared to more strict contour shapes. Additionally, for users interested in analyzing a convective system from its very early initiation stages as a convective cell, the TAMS identification criterion of requiring an embedded 219-K core can sometimes inhibit the detection of these early system stages. However, it's important to mention that users have the flexibility to alter the criteria used in the software's settings.

## 4.2 Development Goals

Our development goals encompass the incorporation of additional settings into our system. These new settings include, but are not limited to:

1. Include other methods for computing CTT from $T_b$.

2. Implement a projection option based on trajectory history.

3. Implement the original TAMS linking method for comparison.

4. Implement a graph-based linking method with pruning, for better handling of (and recording of) both merging and splitting

5. In CE identification, make the application of the convex hull operation optional, and support other shape-simplification options such as buffer.

6. Combine an AEW tracker (Lawton et al., 2022) with TAMS, as detailed in the study by Núñez Ocasio et al. (2020b).

7. Extract the code for reducing the CE dataframe to the MCS dataframe and the reduction from the MCS dataframe to the MCS stats dataframe, so that it can be used in custom workflows outside of `tams.run`

8. More documentation examples, including testing of generated idealized cases

*Code and data availability.* The TAMS open-source code described in this work is publicly available via the following GitHub repository: https://github.com/knubez/TAMS (last access: 28 January 2024). TAMS can be installed using pip or conda/mamba. Jupyter Notebooks and sample MSG and MPAS data can be found in TAMS documentation page: https://tams.readthedocs.io/en/latest/. The version of the code used in this paper is available at https://doi.org/10.5281/zenodo.8393890 (Núñez Ocasio, K. M. and Moon, Z. L., 2023). MSG satellite data for figures related to CPEX-CV can be downloaded via EUMETSAT with a user account: https://eoportal.eumetsat.int.



*Author contributions.* KMNO and ZLM together led the overall development of the TAMS v2.0 code. Both authors contributed to the translation of TAMS v1.0 from MATLAB to TAMS v2.0 in Python while ZLM led the technical aspect of the translation. ZLM led the optimization of the code performance, packaging, and workflow. Both KMNO and ZLM contribute to the analysis, visualization, and writing of this paper.

*Competing interests.* The contact author has declared that none of the authors has any competing interests.

*Acknowledgements.* We thank the reviewers. This work was completed while the first author held an NSF NCAR Advanced Study Program Postdoctoral Fellowship. This material is based upon work supported by the NSF National Center for Atmospheric Research, which is a major facility sponsored by the U.S. National Science Foundation under Cooperative Agreement No. 1852977. We acknowledge the high-performance computing support from Cheyenne (doi: 10.5065/D6RX99HX) and Derecho (https://doi.org/10.5065/qx9a-pg09) through the NSF NCAR Computational and Information Systems Laboratory.



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
