# Peer review of "TAMS: A Tracking, Classifying, and Variable-Assigning Algorithm for Mesoscale Convective Systems in Simulated and Satellite-Derived Datasets"

_EGUsphere, 2024_

## Referee Comment (RC1)

**TAMS: A Tracking, Classifying, and Variable-Assigning Algorithm for Mesoscale Convective Systems in Simulated and Satellite-Derived Datasets**

This paper describes a novel tracking algorithm that was originally developed to track MCSs over Africa, and has been enhanced to a more general tool to track MCSs in large observational and model datasets. The manuscript is well-written and the tracking steps are easy to understand based on the figures presented. Another strength of the paper is that many specific examples are given which help the reader to set the technical features into a scientific context and better understand the implications of different tracking options. An outstanding feature of the TAMS algorithm is its capability to work with unstructured grids, which has become more and more important with the advent of global k-scale models. In addition, TAMS provides the possibility to combine tracked MCS features with any other variable or dataset, which is useful to better understand the processes of convective organization from multiple angles. The tracking tool should therefore be of high interest to the weather and climate research community. I recommend the publication of this manuscript after addressing and clarifying the following minor issues.

**General comments:**

- **Introduction:** While the introduction provides a thorough review on existing tracking algorithms, the motivation of why it is important and useful to track mesoscale convective systems in various datasets could be extended. Following the example applications that are mentioned later in the paper, you could, highlight the importance for both forecasting purposes as well as understanding fundamental climate processes such as interactions of weather systems at different spatial scales. Since TAMS focuses on tracking MCSs, it would also be useful to briefly define what an MCS is from a physical point of view (not in terms of cloud top temperature or precipitation thresholds). This can help readers that are not familiar with this weather system to better understand the choices of thresholds for the classifications lateron in the paper.
- **Background flow**: I am not sure I understand how the zonal projection of cloud elements based on the set background flow parameters is combined with the area-overlapping method. Is this only relevant when there are other overlapping cloud elements, but in the wrong direction? How can users make an informed decision of what background flow parameter to choose?
- **Grid-independence:** Can you define what is meant by grid-independent and what the limitations or minimum requirements are. In particular, it was not quite clear to me what the requirements for the datasets to be matched in the variable-assignment are. Is it only the time dimension that need to fit the timesteps of the track-input data + latitude and longitude information? Does this function also work with unstructured grids or 3D data?
- **Unstructured data:** The polygons in Figure 3 seem to suggest that there are almost no differences between the tracking based on the regridded MPAS data compared to the native-grid MPAS data. Since the reason to track on unstructured data rather than on a regridded version of it is to preserve more information, I am wondering if TAMS can make use of the fine-scale structures from the native grid? Do you expect larger differences in the tracked features when the resolution is even higher (say 4km) or if variables that are noisier than Tb are used as the main tracking field (e.g. precipitation)? A brief discussion of the advantages of tracking on native grids could be useful here.

**Detailed comments**

- L. 11 : remove "robustness" because we are only starting to understand how robust results from objective tracking actually are. And I think the more important point here is that the statistical analysis of MCSs itself has only become possible with the help of objective/automated tracking methods because you can enhance your analysis from case studies to more climatological analyses

- L. 18: Before introducing the overlapping technique as a method to link features over time, it would be useful to briefly explain the concept of feature/object detection. In addition, I suggest to already mention the polygon/convex hull-method that is essential for TAMS here.
- L. 33: Explain "graph node"
- L. 41: Explain TempestExtreme in some more detail, as its flexible design is similar to tobac, but the included algorithms differ. Has TempestExtreme been used to track MCSs?
- L. 58-59: Please provide the reference to the publicly available MCS datasets mentioned here.
- L. 74: Does the cold core only have to appear once during the MCS lifetime?
- L. 85: 3km -> 4km IR data?
- L. 141: Does the *tams.run* function also work with other datasets as precipitation or only if you do the variable-assignment separately?
- L. 143: Can a minimum area criterion be applied in the identification step or only in the classification step?
- L. 162: How is the Matlab function for unstructured grids embedded/implemented in the python framework?
- L. 187: The example cloud elements shown in Figure 4 and the discussion on which tracking parameters lead to a continuous MCS track for the example in the red box are very useful. Can you explain the reason why the setting with u= -15 m/s is needed and why the elements would otherwise not be connected even if they overlap at any point in time?
- Fig. 5: It is an interesting feature that the probability for any MCS area between 10e3 and 10e6 km2 is pretty constant for the default and looser Tb thresholds. In contrast, the stricter Tb threshold option (Tb - 15 K ) results in a distinct peak. Are these characteristics a consequence of the convex hull method or how can these results be interpreted?
- Fig. 6: Although it is clear that darker colors represent higher frequencies, it would be useful to add a colorbar to this figure.
- L. 202: remove one "here"
- L. 201-204: Do you think there is a physical explanation for the different relationships between area and duration for different cold core thresholds or can this feature be attributed to the tracking technique?
- Section 2.3: Can the classes be modified or do you plan to enhance the package such that they can be in the future? It would be a useful tool to allow the users to set definitions for the classes themselves since these could be very dependent on the research questions asked.
- Section 3.3: Maybe you could specify a few examples on the posprocesssing and visualization functionalities that you can find (e.g., plotting spatial maps including the convex hull masks, identified CE numbers and track line overlaid with input data, which statistics can be derived quickly via high-level functions?)
- Section 2.4: In this section, it could be very useful to show an example figure of a contour overlaid with another dataset (e.g. precipitation).
- Section 3.2: Clarify that Yang and Slingo (2001) estimate the infrared brightness temperature which might be more directly related to IR radiances from geostationary satellites than the actual cloud top temperature. In this paragraph, I also suggest to motivate why this functionality is useful, i.e. comparing trackings in model data to satellite observations.

- L. 286-297: These are all great application examples. I suggest to add a subsection (e.g. "Applications") before the summary since this is all new information and it deserves to be highlighted in the main body of the paper.

---

## Editor Comment (EC1)

**Title:** TAMS: A Tracking, Classifying, and Variable-Assigning Algorithm for Mesoscale Convective Systems in Simulated and Satellite-Derived Datasets
**Author:** Kelly M. Núñez Ocasio and Zachary L. Moon
**MS No:** egusphere-2024-259
**MS type:** Methods for assessment of models

**General comments**

This preprint introduces the Tracking Algorithm for Mesoscale Convective Systems (TAMS), highlighting its development, features, and usability. The introduction offers a comprehensive overview of tracking methodologies, setting a strong foundation for understanding TAMS. Originally focused on analyzing mesoscale convective systems (MCSs) over Africa using satellite data, TAMS has evolved into an open-source Python package for tracking and classifying both observed and simulated MCSs. Notable advancements on tracking MCSs algorithm include support for unstructured grids and grid-independent tracking, enhancing its versatility and applicability. The paper meticulously details each algorithmic step, visualization techniques, and post-processing tools, making it accessible and informative. Additionally, it outlines available settings, helper functions, and ongoing development goals, underscoring TAMS' adaptability and broad utility. Overall, the manuscript presents TAMS as a cutting-edge, adaptable, and freely available tracking algorithm, suitable for the "methods for assessment of models" type. Moreover, the TAMS documentation page offers additional details and visual aids to enhance understanding of the tracking process. Overall, the manuscript provides valuable insights into the development and application of TAMS, contributing significantly to the field of MCSs tracking and assessment. With minor revisions and clarifications, it will be well-suited for publication.

**Specific comments**

Line 76: Could you kindly clarify if the 'fixed climatological zonal propagation speed' corresponds to the 'u projection' as discussed in Lines 187 and Table 1? If so, would it be possible to include this abbreviation at Line 76 for clarity? Alternatively, if they are not the same, could you please elaborate on what the 'u projection' refers to at Line 187? Additionally, could you briefly explain how the fixed climatological zonal propagation speed operates, including the atmospheric level used? Should users provide this information, or is it already incorporated into the algorithm? Furthermore, which dataset is utilized to derive this climatological data? Is it based on an average spanning 30 years? Would it be advisable for users to provide average winds for the same period they are tracking the MCSs? Lastly, considering its longitudinal focus, how does the tracking algorithm accommodate regions where the mean flow is latitudinal, such as low-level jet regions along mountainous areas in the USA and South America?

Line 120: It might be beneficial to clarify whether the variable assignment method allows for the identification of the environment in which the MCSs develop, or if its primary function is to assist in defining the MCSs based on various criteria, similar to the approach used in the Prein et al. (2024) study.

Section 2.1 Identification: It would be beneficial to include a concise description of the necessary setups. For example, clarifying whether it is required to provide information regarding the timestep, grid spacing, and structure of the input file could enhance understanding. Additionally, it would be beneficial to discuss why the default thresholds used to define the cloud elements (CE) are the chosed ones and whether users can modify them. Additionally, recommendations for modifications, particularly in scenarios such as tracking MCSs over South America as in Prein et al. (2024), should be provided. Explaining the factors considered when making adjustments and outlining best practices would enhance user understanding.

Line 173: It would be beneficial to provide a brief description of forward linking and recommend its usage under specific circumstances. Additionally, elaborating on the ongoing development of linking methods and their significance would enhance the understanding of their importance. Clarifying whether these methods cater to specific regions or different types of systems would also add valuable insights into their applicability and utility.

Section 2.3 and Table 1: This is another intriguing and positive aspect of TAMS that sets it apart from other algorithms. Further clarification on the definition of MCCs, particularly regarding eccentricity, would be beneficial. Is it essential for the eccentricity to be less than or equal to 0.7 throughout the lifespan of all MCSs to be considered an MCC?

Line 224: The mention of "CCCs, which include squall-line type convection" might cause confusion. It would be beneficial to briefly define "squall line type convection" and clarify why the MCCs in the algorithm cannot be of that type.

In Section 2.4, elaborating on the assignment of variables would enhance understanding of this critical aspect of the tracking algorithm. For example, the mention of computing related statistics (line 234) warrants further explanation. It would be helpful to briefly describe the nature of these statistics—are they averages related to the entire MCSs or to the CE elements? Additionally, it could be beneficial to clarify if users can select the type of statistics to be computed.

The significance of Figure 9 lies in its portrayal of MCS intensity across different categories. However, clarity is needed regarding whether the information pertains to a single moment of the MCS lifecycle (one CE), an individual value within this CE, or an average of the entire lifecycle. Such clarity is crucial for understanding and comparing the intensity of these systems over large timescales and across different regions. Depending on the objective or approach, employing a kernel of pixels to define the most intense region of the CE could be beneficial, drawing inspiration from methodologies outlined by Machado et al. (1998) and Vila et al. (2008). This suggestion holds potential for future implementation within TAMS.

**Technical corrections**

I would kindly suggest to review overall the text in order to not repeat defined terms. For

instance, "tropical cyclones" could be defined in line 69 and not later at line 101. Similarly, "cloud elements" were defined in lines 72-73 and did not need redefinition in line 96. Please review all acronyms throughout the text for consistency. Please review the consistent usage of terms such as IR, TCC, and Tb throughout the manuscript.

Line 22: Please check for typographical ("[...] MCSs. (An MCS [...])") for clarity and consistency.

Line 88: Please double-check the reference to "MCSs Africa" for accuracy.

Line 168: Kindly review for clarity regarding the MPAS data being referenced.

Lines 219 and 223: Please confirm if "MCC" is intended instead of "MCS" in these lines.

Line 131: Please provide the TAMS GitHub access link. This addition will enhance accessibility for readers interested in accessing the algorithm as they are reading.

Line 181: Please clarify the reference to "ID". Specify the term's meaning and relevance within the context of the manuscript.

Figure 7 caption: Please verify the correct year, as it appears to be 2022 instead of 2002. Kindly confirm.

Figure 8 caption: Please include information about the domain used in Figure 8. Is it the same as in Figure 7?

Figure 9 caption: Please add corresponding legends (a) and (b) for clarity. Consider reordering the elements to list area-mean rain rates first, followed by (a), and then the minimum Tb, designated as (b).

Line 260: Please consider reviewing and rephrasing "reaching its 'precipitation rates peak'".

**References**

Machado, L. A. T., Rossow, W. B., Guedes, R. L., & Walker, A. W. (1998). Life Cycle Variations of Mesoscale Convective Systems over the Americas. *Monthly Weather Review*, 126, 1630–1654.
https://doi.org/10.1175/1520-0493(1998)126%3C1630:LCVOMC%3E2.0.CO;2

Prein, A. F., Feng, Z., Fiolleau, T., Moon, Z. L., Núñez Ocasio, K. M., Kukulies, J., Roca, R., Varble, A. C., Rehbein, A., Liu, C., Ikeda, K., Mu, Y., & Rasmussen, R. M. (2024). Km-Scale Simulations of Mesoscale Convective Systems Over South America—A Feature Tracker Intercomparison. *Journal of Geophysical Research: Atmospheres*, 129(8), e2023JD040254.
https://doi.org/10.1029/2023JD040254

Vila, D. A., et al. (2008). Forecast and tracking the evolution of cloud clusters (ForTraCC) using satellite infrared imagery: methodology and validation. *Weather and Forecasting*, 23, 233–245. https://doi.org/10.1175/2007WAF2006121.1

---

## Author Comment (AC1)

**Response to Referees and Editor:**

Each referee and editor's comments will be in black font. Responses will be in this blue font while quoted revisions to the text will be in this orange font, so that the two may be easily distinguished. Responses are separated by individual referees to address each of the comments separately.

We are grateful to the referees and editor for their feedback and very helpful comments. We made considerable changes to the paper.

**RC1: Julia Kukulies**

This paper describes a novel tracking algorithm that was originally developed to track MCSs over Africa, and has been enhanced to a more general tool to track MCSs in large observational and model datasets. The manuscript is well-written and the tracking steps are easy to understand based on the figures presented. Another strength of the paper is that many specific examples are given which help the reader to set the technical features into a scientific context and better understand the implications of different tracking options. An outstanding feature of the TAMS algorithm is its capability to work with unstructured grids, which has become more and more important with the advent of global k-scale models. In addition, TAMS provides the possibility to combine tracked MCS features with any other variable or dataset, which is useful to better understand the processes of convective organization from multiple angles. The tracking tool should therefore be of high interest to the weather and climate research community. I recommend the publication of this manuscript after addressing some minor issues. We thank RC1 for their positive review, we are pleased they appreciate the novelty and contribution of the algorithm to the weather and climate research community.

**General comments:**
● Introduction: While the introduction provides a thorough review on existing tracking algorithms, the motivation of why it is important and useful to track mesoscale convective systems in various datasets could be extended. Following the example applications that are mentioned later in the paper, you could, highlight the importance for both forecasting purposes as well as understanding fundamental climate processes such as interactions of weather systems at different spatial scales. Since TAMS focuses on tracking MCSs, it would also be useful to briefly define what an MCS is from a physical point of view (not in terms of cloud top temperature or precipitation thresholds). This can help readers that are not familiar with this weather system to better understand the choices of thresholds for the classifications later on in the paper. We have now included a new introductory paragraph that defines an MCS and states the importance of understanding these systems as suggested.

● Background flow: I am not sure I understand how the zonal projection of cloud elements based on the set background flow parameters is combined with the area-overlapping method. Is this only relevant when there are other overlapping cloud elements, but in the wrong direction? How can users make an informed decision of what background flow parameter to choose?

The background flow is relevant for when MCSs propagate in an environment with very prominent background flow such as jets for which the MCS may propagate faster than the expected maximum overlap from consecutive images. To make this clear we have now added the following sentences and example to the Tracking section:

Accounting for background flow by projecting the CE before calculating overlap ensures that those MCSs that move fast are being tracked. For example, over Africa, the African easterly jet accounts for a large part of the MCS propagation speed. Thus, depending on the resolution being used the user may want to add a projection of -10 m $s^{-1}$ \citep{NunezOcasio2020TAMS}.

● Grid-independence: Can you define what is meant by grid-independent and what the limitations or minimum requirements are. In particular, it was not quite clear to me what the requirements for the datasets to be matched in the variable-assignment are. Is it only the time dimension that need to fit the timesteps of the track-input data + latitude and longitude information? Does this function also work with unstructured grids or 3D data?
We added statements to the identification and tracking sections to clarify what we mean by grid-independent. We also clarified the coordinate requirements for matching data to CE or MCS polygons ("variable assignment").

Yes, you can use 3-D data, but, with our current tooling, since the grouping is on lat/lon, all vertical levels will be put into the same bucket. Also, our current tooling is meant for only one time, and it is up to the user to provide the correct input. But we hope to expand this in the future, e.g. to auto-parallelize over times.

● Unstructured data: The polygons in Figure 3 seem to suggest that there are almost no differences between the tracking based on the regridded MPAS data compared to the native-grid MPAS data. Since the reason to track on unstructured data rather than on a regridded version of it is to preserve more information, I am wondering if TAMS can make use of the fine-scale structures from the native grid? Do you expect larger differences in the tracked features when the resolution is even higher (say 4km) or if variables that are noisier than Tb are used as the main tracking field (e.g. precipitation)? A brief discussion of the advantages of tracking on native grids could be useful here.
It is possible, though currently somewhat difficult, to track on the exact contour CEs instead of their convex hulls in the TAMS framework. In this case, the increased detail shown in Figure 3a would be preserved. We plan to make this easier in the future, as well as adding other shape-simplification methods less "extreme" than convex hull.

We note that in Figure 3b around 100 E, there is a pretty significant difference in the convex hulls, with the higher-resolution native data leading to a separate CE around 5 N, while the coarsened regridded data has this as part of the larger CE. We added this to the text along with a brief statement about the advantage of the native.

**Detailed comments**

L. 11 : remove "robustness" because we are only starting to understand how robust results from objective tracking actually are. And I think the more important point here is that the statistical analysis of MCSs itself has only become possible with the help of objective/automated tracking methods because you can enhance your analysis from case studies to more climatological analyses.
Done and we have added this statement to the text. Thank you.

L. 18: Before introducing the overlapping technique as a method to link features over time, it would be useful to briefly explain the concept of feature/object detection. In addition, I suggest to already mention the polygon/convex hull-method that is essential for TAMS here.
We have addressed and added to the introduction.

L. 33: Explain "graph node"
We removed the term to use 'graph algorithm' insead.

L. 41: Explain TempestExtreme in some more detail, as its flexible design is similar to tobac, but the included algorithms differ. Has TempestExtreme been used to track MCSs?
We couldn't find a case of TempestExtreme being used on MCss. We have made this clear that they track TCs and extratropical cyclones. .

L. 58-59: Please provide the reference to the publicly available MCS datasets mentioned here.
Done.

L. 74: Does the cold core only have to appear once during the MCS lifetime?
No, it always has to have a cold core, otherwise it is not identified as a potential MCS candidate. We have added this in the text for clarity.

In TAMS v2.0 the user can disable this within the TAMS code, for example just setting the cold core threshold to be the same as the cloud threshold.

L. 85: 3km -> 4km IR data?
For is a better estimate, thanks.

L. 141: Does the tams.run function also work with other datasets as precipitation or only if you do the variable-assignment separately?
Currently it uses the variable called "pr", but this could technically be something other than precipitation.

L. 143: Can a minimum area criterion be applied in the identification step or only in the classification step?
A minimum area criterion is used in both identification and classification steps, but they are different. In identification, this can be disabled.

L. 162: How is the Matlab function for unstructured grids embedded/implemented in the python framework?
Assuming the reviewer means Matplotlib, Matplotlib's tricontourf function is called to generate contour definitions, which are then transformed into Shapely polygons.

L. 187: The example cloud elements shown in Figure 4 and the discussion on which tracking parameters lead to a continuous MCS track for the example in the red box are very useful. Can you explain the reason why the setting with u= -15 m/s is needed and why the elements would otherwise not be connected even if they overlap at any point in time?
The overlap is not enough to meet the threshold without the projection.

Fig. 5: It is an interesting feature that the probability for any MCS area between 10e3 and 10e6 km2 is pretty constant for the default and looser Tb thresholds. In contrast, the stricter Tb threshold option (Tb - 15 K ) results in a distinct peak. Are these characteristics a consequence of the convex hull method or how can these results be interpreted?
We do not think this result is due to the convex hull method but rather to the more diverse changes in the area of clouds if choosing a warmer temperature compared to the colder region which seems to have a shorter range of preferred areas.

For example, 235 - 15 = 220 K is essentially a cold-core threshold. The result indicates the cold cores have more of a "preferred" lifetime-maximum area (10^5) than general cloud shields which seem to have a more uniform distribution. Cold cores can only get so big, and we are looking at the maximum area over the lifetime, giving rise to this distribution shape. We find our col core area numbers are similar to those from PyFLEXTRKR, we have added this reference.

Fig. 6: Although it is clear that darker colors represent higher frequencies, it would be useful to add a colorbar to this figure.
We added a colorbar with more and matched contour levels to this figure.

L. 202: remove one "here"
Thanks

L. 201-204: Do you think there is a physical explanation for the different relationships between area and duration for different cold core thresholds or can this feature be attributed to the tracking technique?
Could be related to both tracking techniques but also physical processes, for example those related to cold pool dynamics. We have made this point in the text.

Section 2.3: Can the classes be modified or do you plan to enhance the package such that they can be in the future? It would be a useful tool to allow the users to set definitions for the classes themselves since these could be very dependent on the research questions asked.
We have done custom classification before for previous and current inter-comparison projects. It is not too difficult to set up TAMS's data is in a dataframe type that is pretty easy to work with.

But we may consider parameterizing the official classification in the future, we appreciate the suggestion.

Section 3.3: Maybe you could specify a few examples on the postprocesssing and visualization functionalities that you can find (e.g., plotting spatial maps including the convex hull masks, identified CE numbers and track line overlaid with input data, which statistics can be derived quickly via high-level functions?)
We have added these comments to the section, thanks.

Section 2.4: In this section, it could be very useful to show an example figure of a contour overlaid with another dataset (e.g. precipitation).
Thank you, we have included a new figure (new Figure 9) that is a visual example of the variable assignment demonstrating that the geospatial selection also works for unstructured-grid data.

Section 3.2: Clarify that Yang and Slingo (2001) estimate the infrared brightness temperature which might be more directly related to IR radiances from geostationary satellites than the actual cloud top temperature. In this paragraph, I also suggest to motivate why this functionality is useful, i.e. comparing trackings in model data to satellite observations.
We have modified this section to clarify that these are only estimates of the CTT.

L. 286-297: These are all great application examples. I suggest to add a subsection (e.g. "Applications") before the summary since this is all new information and it deserves to be highlighted in the main body of the paper.
Done, thanks.

**RC2: Anonymous Referee**

The manuscript introduces the Python-based version of the Tracking Algorithm for Mesoscale Convective Systems (TAMS), TAMS v2.0. In addition to describing the algorithm, the authors provide some examples to help readers understand it. The topic is within the scope of GMD, and the tool is helpful in the scientific community. However, the manuscript has three significant limitations and cannot be published in its current form.
We thank the reviewer for the suggestions and comments.

**Major comments**
1. The algorithm introduction involves too many software functions (e.g., tams.identify, ctt_core_threshold, etc.) but lacks a scientific description. It makes the manuscript look more like a user guide or technical documentation than a scientific paper. I strongly suggest the authors rewrite the algorithm's description in a more general way: tell the readers the principles of the algorithm but not list those built-in functions. For example, for ctt_core_threshold, please tell the readers the relevant physical variable but not this type of acronym used in the tool.

We appreciate this suggestion. We reduced our usage of names of functions and names in code and such to a few key functions that we expect not to change in the future.

2. I don't find enough updates in TAMS v2.0 compared to TAMS v1.0. However, since TAMS v2.0 is a Python-based and open-source software and TAMS v1.0 has not been published independently, describing TAMS v2.0 in a separate paper is acceptable. I suggest the authors avoid describing TAMS v1.0 too much or remove the version label entirely. The Python-based TAMS can be an independent tool.

We disagree as we do believe TAMS v2.0 is a significant update from the initial version providing the user with helper grid-independent tracking, parallelization, variable assignment (not just precipitation), and the rest of the updates we detail in this paper. However, to not confuse the reader and per your suggestion, we have removed the version label and just called the first version 'original TAMS' and the new one 'TAMS'.

3. The tool is designed to track MCSs. However, the tracked MCSs contain systems lasting only a few hours. Are they really MCSs? I think the tool tracks all convective systems but not just MCSs. Please clarify the terms throughout the manuscript.

In the TAMS framework, disorganized short-lived (DSL) systems are still considered MCSs since they have sufficiently large cold-core areas at all times. There is no lower bound on the duration for this type.

**Minor comments**
Lines 33: How about "identifies and tracks cloud clusters using IR and a corresponding graph node via the area overlap method"?
Fixed.

Line 67: "However, before discussing the Python-based TAMS, we will review the first version of TAMS (TAMS v1.0), which is written in MATLAB"?
Fixed.

Lines 86-87: I'm afraid I have to disagree that using IMERG is an advantage of TAMS. The precipitation assignment is the last step of TAMS v1.0, which doesn't affect the tracking algorithm. To my knowledge, FLEXTRKR used Stage IV over the United States, which had a resolution of 4 km, and IMERG for the global MCS dataset.

We disagree with this reviewer comment as back in 2020 TAMS was indeed the first objective tracker to provide combined IMERG and MCS tracks. We tried to find the reference the reviewer is referring to and we find Li et al. 2021 to be the closest. NCEP Stage IV seems to be a similar but separate product from IMERG and even then, this paper was published in 2021, still making our statement valid.
Note that IMERG is not required for TAMS; another precipitation dataset could be used.

Line 88: "MCSs in Africa"?
Thanks

Line 104: Delete "was written in MATLAB, and it"?
We would like to keep this statement.

Line 113: Delete the first sentence.
We would like to keep this sentence.

Line 156: Correct the citation format and the full name of IFS.
Done.

Line 157: Correct the citation format.
Done.

Line 168: "from the regridded MPAS data"?
Fixed.

Line 170: What do you mean by current and previous CTTs?
We changed "CTT" to use the term "time" instead of "CTT".

Lines 187-188: What do you mean here?
Cases 3 and 4 of Table 1. We have clarified in the text.

Figure 5 caption: Correct the description for panels in different rows and columns. Similar to Figure 6. Please add a legend for the color shading in Figure 6.
We believe the description of Figure 5 is clear. We have added a colorbar to Figure 6.

Lines 208-211: What is the minimum duration that an MCS can have? Can a track be considered an MCS if it lasts only 1 hour? Or do you only consider MCCs and CCCs as MCSs? Please rewrite these confusing sentences.
These criteria all evaluated for all the MCSs that were identified and tracked. In theory, if a CE is identified as a potential MCS at only one time because it satisfies the identification criteria, then yes, the shortest lifetime for an MCS could be an hour. However, this would be very unlikely or rare.

Line 225: If so, can they still be considered MCSs?
Yes, because of the criteria used to identify an MCS in the identification step.

**RC3: Anonymous Referee**

**General comments**
This preprint introduces the Tracking Algorithm for Mesoscale Convective Systems (TAMS), highlighting its development, features, and usability. The introduction offers a comprehensive overview of tracking methodologies, setting a strong foundation for understanding TAMS. Originally focused on analyzing mesoscale convective systems (MCSs) over Africa using satellite data, TAMS has evolved into an open-source Python package for tracking and

classifying both observed and simulated MCSs. Notable advancements on tracking MCSs algorithm include support for unstructured grids and grid-independent tracking, enhancing its versatility and applicability. The paper meticulously details each algorithmic step, visualization techniques, and post-processing tools, making it accessible and informative. Additionally, it outlines available settings, helper functions, and ongoing development goals, underscoring TAMS' adaptability and broad utility. Overall, the manuscript presents TAMS as a cutting-edge, adaptable, and freely available tracking algorithm, suitable for the "methods for assessment of models" type. Moreover, the TAMS documentation page offers additional details and visual aids to enhance understanding of the tracking process. Overall, the manuscript provides valuable insights into the development and application of TAMS, contributing significantly to the field of MCSs tracking and assessment. With minor revisions and clarifications, it will be well-suited for publication.

We appreciate RC3's positive review and are delighted that they recognize the novelty and contribution of the MCS tracking community.

**Specific comments**
Line 76: Could you kindly clarify if the 'fixed climatological zonal propagation speed' corresponds to the 'u projection' as discussed in Lines 187 and Table 1? If so, would it be possible to include this abbreviation at Line 76 for clarity? Alternatively, if they are not the same, could you please elaborate on what the 'u projection' refers to at Line 187? Additionally, could you briefly explain how the fixed climatological zonal propagation speed operates, including the atmospheric level used? Should users provide this information, or is it already incorporated into the algorithm? Furthermore, which dataset is utilized to derive this climatological data? Is it based on an average spanning 30 years? Would it be advisable for users to provide average winds for the same period they are tracking the MCSs? Lastly, considering its longitudinal focus, how does the tracking algorithm accommodate regions where the mean flow is latitudinal, such as low-level jet regions along mountainous areas in the USA and South America?

Currently, the user must provide the fixed u projection, with the default being 0. This is used to translate polygons from the previous time step before computing overlap with the current time. The user could estimate what value to use based on a climatology of wind data but we leave it to the user's discretion to decide on the specific value to choose.

The current projection method works well for the application to African MCSs, but we agree it may not be very useful for regions such as South America. We hope in the future to provide the ability to specify a 2-D field of projection velocity (u and v).

Line 120: It might be beneficial to clarify whether the variable assignment method allows for the identification of the environment in which the MCSs develop, or if its primary function is to assist in defining the MCSs based on various criteria, similar to the approach used in the Prein et al. (2024) study.

We updated the text to cite how we used this for the Prein et al. (2024) study.

Section 2.1 Identification: It would be beneficial to include a concise description of the

necessary setups. For example, clarifying whether it is required to provide information regarding the timestep, grid spacing, and structure of the input file could enhance understanding. Additionally, it would be beneficial to discuss why the default thresholds used to define the cloud elements (CE) are the chosed ones and whether users can modify them. Additionally, recommendations for modifications, particularly in scenarios such as tracking MCSs over South America as in Prein et al. (2024), should be provided. Explaining the factors considered when making adjustments and outlining best practices would enhance user understanding.

The user does not need to provide timestep and grid spacing information to TAMS but they should be aware of it when considering overlap threshold and the addition of cloud projection for the tracking step. The data provided for the identification functionality should be xarray format. We specify in the text that the identification criteria follow the original TAMS but we have now added the statement that identification criteria can be altered by the user as we did in Prein et al 2024 following your previous suggestion.

Line 173: It would be beneficial to provide a brief description of forward linking and recommend its usage under specific circumstances. Additionally, elaborating on the ongoing development of linking methods and their significance would enhance the understanding of their importance. Clarifying whether these methods cater to specific regions or different types of systems would also add valuable insights into their applicability and utility.

Forward linking is a term we use to describe the direction of the overlapping method. The overlapping method is introduced and discussed in the introduction and description of tracking.

Section 2.3 and Table 1: This is another intriguing and positive aspect of TAMS that sets it apart from other algorithms. Further clarification on the definition of MCCs, particularly regarding eccentricity, would be beneficial. Is it essential for the eccentricity to be less than or equal to 0.7 throughout the lifespan of all MCSs to be considered an MCC?

The criteria including the eccentricity criterion need not to be met for consecutive times but for at least for 6 hours within the lifetime of the system. We have now added that MCC as defined here follow's Maddox definition of a Mesoscale Convective Complex.

Line 224: The mention of "CCCs, which include squall-line type convection" might cause confusion. It would be beneficial to briefly define "squall line type convection" and clarify why the MCCs in the algorithm cannot be of that type.

We have made this clarification now  in the text, thank you.

In Section 2.4, elaborating on the assignment of variables would enhance understanding of this critical aspect of the tracking algorithm. For example, the mention of computing related statistics (line 234) warrants further explanation. It would be helpful to briefly describe the nature of these statistics—are they averages related to the entire MCSs or to the CE elements? Additionally, it could be beneficial to clarify if users can select the type of statistics to be computed.

Over MCSs, officially, but users can also apply the tooling to compute stats over the CEs (cloud elements).This is now described in the text as well as the mention of the default statistics: mean and standard deviation.

If \texttt{tams.run} is used instead of running each step separately, the statistics will also include, mean and standard deviation of the cold cores and precipitation rates.

The significance of Figure 9 lies in its portrayal of MCS intensity across different categories. However, clarity is needed regarding whether the information pertains to a single moment of the MCS lifecycle (one CE), an individual value within this CE, or an average of the entire lifecycle. Such clarity is crucial for understanding and comparing the intensity of these systems over large timescales and across different regions. Depending on the objective or approach, employing a kernel of pixels to define the most intense region of the CE could be beneficial, drawing inspiration from methodologies outlined by Machado et al. (1998) and Vila et al. (2008). This suggestion holds potential for future implementation within TAMS. We have now clarified that these are statistics that include each time step of the MCSs, thus we combined the CEs before creating the plot. Thank you very much for the reference, we have taken note of this fur future implementation.

**Technical corrections**
I would kindly suggest to review overall the text in order to not repeat defined terms. For instance, "tropical cyclones" could be defined in line 69 and not later at line 101. Similarly, "cloud elements" were defined in lines 72-73 and did not need redefinition in line 96. Please review all acronyms throughout the text for consistency. Please review the consistent usage of terms such as IR, TCC, and Tb throughout the manuscript.
We have reviewed the terms for consistency.

Line 22: Please check for typographical ("[...] MCSs. (An MCS [...])") for clarity and Consistency.
We checked for consistency for "an MCS" (desired) vs "a MCS" (not). Line 22 does not have, however, terms 'MCS' or "MCSs".

Line 88: Please double-check the reference to "MCSs Africa" for accuracy.
Fixed, we meant "MCSs in Africa".

Line 168: Kindly review for clarity regarding the MPAS data being referenced.
Fixed, thank you.

Lines 219 and 223: Please confirm if "MCC" is intended instead of "MCS" in these lines.
MCC was intended, fixed. Thanks.

Line 131: Please provide the TAMS GitHub access link. This addition will enhance accessibility for readers interested in accessing the algorithm as they are reading.
The TAMS documentation page link is included at the beginning of this paragraph.

Line 181: Please clarify the reference to "ID". Specify the term's meaning and relevance within the context of the manuscript.

We were just referring to the MCS. This is the only time we used the term 'ID' so we removed it for clarity.

Figure 7 caption: Please verify the correct year, as it appears to be 2022 instead of 2002. Kindly confirm.

Thanks, 2022 is correct.

Figure 8 caption: Please include information about the domain used in Figure 8. Is it the same as in Figure 7?

We modified the caption to clarify that it is the same domain.

Figure 9 caption: Please add corresponding legends (a) and (b) for clarity. Consider reordering the elements to list area-mean rain rates first, followed by (a), and then the minimum Tb, designated as (b).

Noted, but given the y-axis explicit definition of the variable, we don't think a legend is necessary. We have edited the caption description for clarification.

Line 260: Please consider reviewing and rephrasing "reaching its 'precipitation rates peak'".

Done.

**References**

Machado, L. A. T., Rossow, W. B., Guedes, R. L., & Walker, A. W. (1998). Life Cycle Variations of Mesoscale Convective Systems over the Americas. Monthly Weather Review, 126, 1630–1654. https://doi.org/10.1175/1520-0493(1998)126%3C1630:LCVOMC%3E2.0.CO;2

Prein, A. F., Feng, Z., Fiolleau, T., Moon, Z. L., Núñez Ocasio, K. M., Kukulies, J., Roca, R., Varble, A. C., Rehbein, A., Liu, C., Ikeda, K., Mu, Y., & Rasmussen, R. M. (2024). Km-Scale Simulations of Mesoscale Convective Systems Over South America—A Feature Tracker Intercomparison. Journal of Geophysical Research: Atmospheres, 129(8), e2023JD040254. https://doi.org/10.1029/2023JD040254

Vila, D. A., et al. (2008). Forecast and tracking the evolution of cloud clusters (ForTraCC) using satellite infrared imagery: methodology and validation. Weather and Forecasting, 23, 233–245. https://doi.org/10.1175/2007WAF2006121.1

Thank you for the references.

**Response to Editor Peter Caldwell comment:**

We have now revised the text and changes are sufficient now to address the editors' concerns.